# Testing Evapotranspiration Estimates Based on MODIS Satellite Data in the Assessment of the Groundwater Recharge of Karst Aquifers in Southern Italy

**Giovanni Ruggieri [1], Vincenzo Allocca [1,\*], Flavio Borfecchia [2], Delia Cusano [1], Palmira Marsiglia [1]** and **Pantaleone De Vita [1]**

[1] Department of Earth Science, Environment and Resources, University of Naples Federico II, 80126 Naples, Italy; gioruggieri@gmail.com (G.R.); delia.cusano@unina.it (D.C.); palmira.marsiglia@unina.it (P.M.); pantaleone.devita@unina.it (P.D.V.)

[2] Territorial and Production Systems Sustainability Department, Italian National Agency for New Technologies, Energy and Sustainable Economic Development, 00196 Rome, Italy; borfecchia@casaccia.enea.it

\* Correspondence: vincenzo.allocca@unina.it

**Abstract:** In many Italian regions, and particularly in southern Italy, karst aquifers are the main sources of drinking water and play a crucial role in the socio-economic development of the territory. Hence, estimating the groundwater recharge of these aquifers is a fundamental task for the proper management of water resources, while also considering the impacts of climate changes. In the southern Apennines, the assessment of hydrological parameters that is needed for the estimation of groundwater recharge is a challenging issue, especially for the spatial and temporal inhomogeneity of networks of rain and air temperature stations, as well as the variable geomorphological features and land use across mountainous karst areas. In such a framework, the integration of terrestrial and remotely sensed data is a promising approach to limit these uncertainties. In this research, estimations of actual evapotranspiration and groundwater recharge using remotely sensed data gathered by the Moderate Resolution Imaging Spectrometer (MODIS) satellite in the period 2000–2014 are shown for karst aquifers of the southern Apennines. To assess the uncertainties affecting conventional methods based on empirical formulas, the values estimated by the MODIS dataset were compared with those calculated by Coutagne, Turc, and Thornthwaite classical empirical formulas, which were based on the recordings of meteorological stations. The annual rainfall time series of 266 rain gauges and 150 air temperature stations, recorded using meteorological networks managed by public agencies in the period 2000–2014, were considered for reconstructing the regional distributed models of actual evapotranspiration (AET) and groundwater recharge. Considering the MODIS AET, the mean annual groundwater recharge for karst aquifers was estimated to be about 448 mm·year$^{-1}$. In contrast, using the Turc, Coutagne, and Thornthwaite methods, it was estimated as being 494, 533, and 437 mm·year$^{-1}$, respectively. The obtained results open a new methodological perspective for the assessment of the groundwater recharge of karst aquifers at the regional and mean annual scales, allowing for limiting uncertainties and taking into account a spatial resolution greater than that of the existing meteorological networks. Among the most relevant results obtained via the comparison of classical approaches used for estimating evapotranspiration is the good matching of the actual evapotranspiration estimated using MODIS data with the potential evapotranspiration estimated using the Thornthwaite formula. This result was considered linked to the availability of soil moisture for the evapotranspiration demand due to the relevant precipitation in the area, the general occurrence of soils covering karst aquifers, and the dense vegetation.

**Keywords:** groundwater recharge; karst aquifer; evapotranspiration; remote sensing; MODIS satellite; southern Italy

## 1. Introduction

Groundwater resources of karst aquifers are of fundamental relevance worldwide for human and agricultural water supplies and for sustaining fluvial ecosystems with great geo- and biodiversities [1]. These aquifers are present in many Italian regions where they constitute the main sources of drinking water, playing a strategic role for socio-economic development of the territory, and also the control of bio-geomorphological conditions of groundwater-dependent ecosystems.

Estimating the groundwater recharge of these aquifers, from local to regional spatial scales and from episodic to annual time scales [2–4], is a fundamental tool for the management of groundwater resources, which also involves considering the possible assessment of the effects of climate changes on groundwater recharge [5,6].

In the southern Apennines, karst aquifers have a significant extension, covering about 44% of the whole territory (8560 km$^2$), and form the principal mountain ranges over which the highest orographic precipitations are concentrated. Due to the mainly mountainous features of karst aquifers, the estimation of groundwater recharge is a challenging issue to be tackled due to the lack of high-altitude rain and air temperature gauges, as well as the spatial and temporal discontinuity of recordings [2]. In such a framework, the integration of terrestrial and remotely sensed data is a promising approach to limit these uncertainties and to account for variable land use in karst areas controlling the evapotranspiration demand.

A large part of remote sensing methods has addressed the estimation of soil moisture and its implementation in soil water balance and related groundwater recharge assessment [7].

Specifically, microwave remote sensing techniques are used to map soil moisture and to monitor its temporal dynamics at the regional scale, with an effectiveness that is not matched by other field techniques. This approach is particularly relevant in arid and semiarid regions, where evapotranspiration is usually a very significant component of the water-balance equation [8].

Passive and active methods are used in microwave techniques. In passive methods, the natural thermal emission of the land surface is measured through very sensitive sensors. In active methods, or radar, a microwave signal is emitted and received by the source. The power of the received signal is compared to that emitted to determine the backscattering coefficient. These measurements can be made at any time of the day or night because they are not dependent on solar illumination.

Beyond the soil moisture assessment, in the last few decades, increasing attention has been paid to characterizing moisture fluxes between the ground and lower atmosphere due to evapotranspiration and their effects on water balance using remotely sensed data. To support such an effort are the facts that evapotranspiration is the second-most important factor affecting the terrestrial water budget after precipitation and remote sensing is the only feasible approach to assess it at regional or continental scales. In such a view, the estimation of evapotranspiration using remotely sensed data will be even more important due to its expected increase caused by global warming [9,10].

At this scope, different methods have been developed to estimate evapotranspiration from remote sensing data, varying from empirical approaches to complex methods based on the assimilation of remote sensing data and their coupling with soil–vegetation–atmosphere transfer models (SVAT). The increasing effort in applying remote sensing techniques has been boosted by the inapplicability of classical methods at the regional or continental scales, which are normally used to measure evapotranspiration at the field scale (Bowen ratio, eddy correlation system, soil water balance).

In this framework, remote sensing data with an increased spatial resolution represent a useful tool to estimate evapotranspiration at various temporal and spatial scales. Depending on the methods used, four groups of methods can be recognized [11]:

(a)   Empirical direct methods to estimate evapotranspiration based on the processing of remotely sensed data using semi-empirical models, such as simplified relationships using thermal infrared (TIR) remotely sensed data and meteorological models.

(b)   Residual methods of the energy budget, which use remote sensing data by combining empirical relationships and physical models (such as Surface Energy Balance Algorithm for Land–SEBAL, Simplified Surface Energy Balance Index–S-SEBI) and applying them to evapotranspiration [12–14].

(c)   Deterministic methods, which are based on SVAT models, estimating the different components of the energy budget (Interactions between the Soil Biosphere and Atmosphere–ISBA, Non-Hydrostatic Mesoscale atmospheric model–Meso-NH) and using remote sensing data at different levels, either as input parameters or in data assimilation procedures.

(d)   Vegetation index methods, which are based on the use of remote sensing to compute a reduction factor (such as Kc or Priestley Taylor $\alpha$ parameters) for the estimation of evapotranspiration in comparison to field measurements [15–17].

These approaches are not mutually exclusive due to the possibility of a mutual integration by means of calibrations based on ground measurements at plot scales [18–20].

Regarding advances in remote sensing techniques that are applied to the estimation of actual evapotranspiration [20], Advanced Spaceborne Thermal Emission and Reflection Radiometer (ASTER) and Moderate Resolution Imaging Spectrometer (MODIS) sensor systems, which are both mounted onboard the Terra satellite (EOS AM-1), have to be mentioned due to the improvement of coverage in the TIR bands and spatial resolution in the visible and near-infrared (NIR) bands in comparison to earlier sensor systems [21]. ASTER has a resolution of 15 m in the visible and NIR bands and 90 m in the TIR bands, with a return time of 16 days. The MODIS sensor provides an almost daily frequency of detection with pixel resolutions of 250 and 500 m in the visible and NIR bands, respectively. MODIS also detects land surface temperatures (LSTs) with a potential accuracy of 1.0 °C and pixel width from 1 to 5 km.

Different attempts to validate actual evapotranspiration from MODIS were carried out, as in the case of the comparison with measurements of eddy covariance flux that were collected by the Soil Moisture Atmospheric Coupling Experiment (SMACEX) over the Walnut Creek watershed in Iowa in 2012 [22].

In this research, for the first time in southern Italy, the groundwater recharge of karst aquifers was assessed using the integration of ground measurements gathered by meteorological networks, with estimations of actual evapotranspiration derived from the MODIS satellite data, which were aggregated at the mean annual time scale for the period 2000–2014. Moreover, estimations of actual evapotranspiration made by MODIS actual evapotranspiration (MODIS AET) data were compared to results of classical approaches applied to the estimation of the AET (Coutagne and Turc formulas) and potential evapotranspiration (PET) (Thornthwaite formula).

The proposed approach is consistent with the even more increasing application of remotely sensed data to hydrological sciences, which is fostered by the great availability of qualitative and quantitative information at large scales, from regional to continental [23].

## 2. Hydrogeological and Climatic Settings

The southern Apennines consist of a series of mountain ranges in which karst aquifers form the principal massifs hosting major groundwater resources [2]. In the study area, stretching over seven regions (Lazio, Abruzzo, Molise, Campania, Basilicata, Puglia, and Calabria), 40 principal karst aquifers with an autonomous groundwater circulation can be recognized (Table 1, Figure 1). These aquifers are mainly formed by Mesozoic carbonate series deposited in paleo-environments of a carbonate platform and varying in lithology from dolomite (Triassic–Liassic) to limestone (Jurassic–Cretaceous) to marly limestones (Paleogene). These series were tectonically deformed and piled up in the fold-and-thrust belt Apennine structure during the Miocene orogenic phases, which were generated by the collision between the African and European plates. After the orogenesis, during Pliocene and Quaternary, extensional tectonic phases took place, resulting in brittle deformation of the rock masses with the development of normal fault systems and pervasive fractur-

ing, which favored the growth of karst phenomena, especially in limestone series. As a consequence, the latter lithologies are generally characterized by the highest permeability grade [24].

**Table 1.** Basic physiographic features and estimations of the annual groundwater recharge coefficient (AGRC) and AGRC for sloped areas (AGRC$_S$) for karst aquifers of the study area (Figure 1) [2].

| ID | Karst Aquifer | Area (km$^2$) | Limestone Area (%) | Summit Plateau and Endorheic Area (%) | AGRC (%) | AGRC$_S$ (%) | Average Altitude (m a.s.l.) |
|----|---------------|---------------|--------------------|---------------------------------------|----------|--------------|-----------------------------|
| 1 | Cerella | 137 | 100 | 0 | 56 | 56 | 655 |
| 2 | Simbruini | 1076 | 94 | 12 | 62 | 57 | 952 |
| 3 | Cornacchia | 723 | 90 | 7 | 59 | 56 | 1324 |
| 4 | Marsicano | 204 | 94 | 5 | 58 | 56 | 1575 |
| 5 | Genzana | 277 | 10 | 34 | 66 | 49 | 1528 |
| 6 | Rotella | 40 | 100 | 40 | 77 | 62 | 1499 |
| 7 | Porrara | 64 | 100 | 25 | 69 | 59 | 1420 |
| 8 | Lepini | 483 | 100 | 2 | 57 | 57 | 617 |
| 9 | Colli Campanari | 97 | 0 | 12 | 54 | 48 | 863 |
| 10 | Capraro | 61 | 0 | 5 | 51 | 48 | 1114 |
| 11 | Campo | 16 | 0 | 13 | 55 | 48 | 1314 |
| 12 | Circeo | 7 | 0 | 0 | 48 | 48 | 163 |
| 13 | Ausoni | 826 | 99 | 15 | 64 | 58 | 607 |
| 14 | Venafro | 365 | 74 | 11 | 60 | 55 | 654 |
| 15 | Totila | 195 | 0 | 8 | 52 | 48 | 940 |
| 16 | Maio | 93 | 98 | 12 | 63 | 58 | 327 |
| 17 | Matese | 588 | 71 | 19 | 64 | 56 | 955 |
| 18 | Tre Confini | 28 | 0 | 4 | 50 | 48 | 913 |
| 19 | Moschiaturo | 85 | 0 | 7 | 51 | 48 | 865 |
| 20 | Massico | 29 | 89 | 0 | 55 | 55 | 334 |
| 21 | Maggiore | 173 | 99 | 0 | 56 | 56 | 344 |
| 22 | Camposauro | 50 | 99 | 4 | 58 | 56 | 807 |
| 23 | Tifatini | 65 | 90 | 2 | 56 | 56 | 257 |
| 24 | Taburno | 43 | 81 | 4 | 57 | 55 | 829 |
| 25 | Durazzano | 52 | 100 | 0 | 56 | 56 | 395 |
| 26 | Avella | 334 | 100 | 9 | 61 | 57 | 617 |
| 27 | Terminio | 167 | 100 | 43 | 78 | 62 | 934 |
| 28 | Capri | 9 | 93 | 0 | 56 | 56 | 152 |
| 29 | Lattari | 245 | 75 | 0 | 54 | 54 | 494 |
| 30 | Salerno | 46 | 13 | 0 | 49 | 49 | 362 |
| 31 | Accellica | 206 | 33 | 0 | 51 | 51 | 689 |
| 32 | Cervialto | 129 | 98 | 20 | 67 | 58 | 1119 |
| 33 | Polveracchio | 114 | 81 | 0 | 55 | 55 | 930 |
| 34 | Marzano | 308 | 97 | 13 | 63 | 57 | 808 |
| 35 | Alburni | 254 | 99 | 42 | 78 | 62 | 917 |
| 36 | Cervati | 318 | 81 | 13 | 62 | 56 | 862 |
| 37 | Motola | 52 | 100 | 4 | 59 | 57 | 1004 |
| 38 | Maddalena | 300 | 59 | 21 | 64 | 54 | 939 |
| 39 | Forcella | 217 | 86 | 5 | 58 | 56 | 676 |
| 40 | Bulgheria | 101 | 68 | 1 | 54 | 54 | 396 |

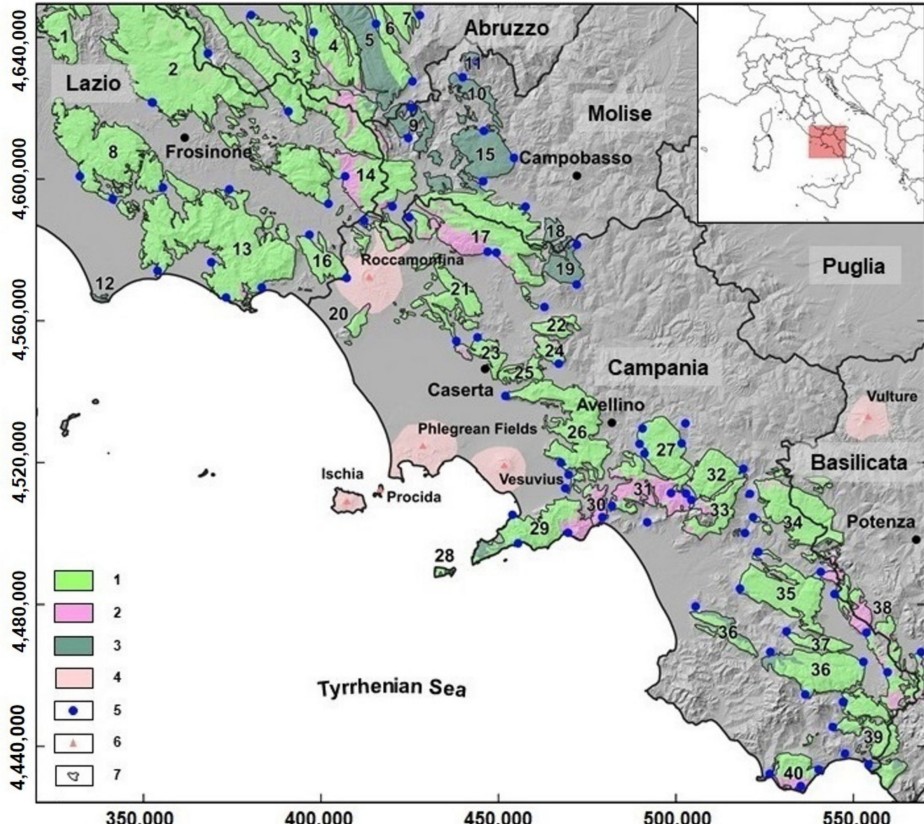

**Figure 1.** Map of the karst aquifers of the study area. Key to symbols: (1) limestone and dolomitic limestone hydrogeological units of a carbonate platform series (Jurassic–Paleogene); (2) dolomitic hydrogeological units of a carbonate platform series (Triassic–Liassic); (3) calcareous-marly hydrogeological units of an outer basin series (Triassic–Paleogene); (4) volcanic centers (Pliocene–Quaternary); (5) main basal springs of karst aquifers; (6) volcanoes; (7) regional boundaries.

Carbonate mountains forming the karst aquifers of southern Italy are usually characterized by summit plateau and endorheic zones due to structural settings and karst development, and by structurally controlled slopes that are related to the morphological evolution of original fault line scarps, with slope angles generally ranging between 30 and 35°, and locally reaching very steep conditions [2,3,25].

In the following Table 1, the main morphological, hydrological, and hydrogeological features of karst aquifers of southern Italy are reported.

A descriptive statistical analysis of the altitude of carbonate mountains forming karst aquifers of southern Italy, distinguished based on their lithological type, is given in the following Figure 2.

The general high permeability grade due to fracturing and karsts, along with summit extended endorheic morphologies, lead to a high groundwater recharge rate that varies between 48 and 78% of the mean annual effective precipitation [2,3], depending on the relative abundance of limestone and dolomite lithologies (Table 1).

A singular features of karst aquifers of the Campania region, particularly of those surrounding volcanic centers of Phlegraean Fields and Somma-Vesuvius, is the covering of thick ash-fall pyroclastic soils that erupted during the Quaternary, whose occurrence strongly controls the annual evapotranspiration rate and regime by means of the development of a dense vegetation cover and groundwater recharge [26–29].

Groundwater circulation emerges mostly in main basal springs (Figure 1), with a mean annual discharge varying from 0.1 to 5.5 $m^3 s^{-1}$ and a perennial regime, which are generally located at the lowest points along the boundaries separating the karst aquifers and the surrounding low-permeability flysch aquitards or aquicludes [30]. Besides the

principal basal one, a minor perched groundwater circulation related to local stratigraphic, structural, and karst factors can occur at a higher altitude across the massifs, feeding springs that are characterized by discharge rates lower than the basal ones, and variable regimes [31].

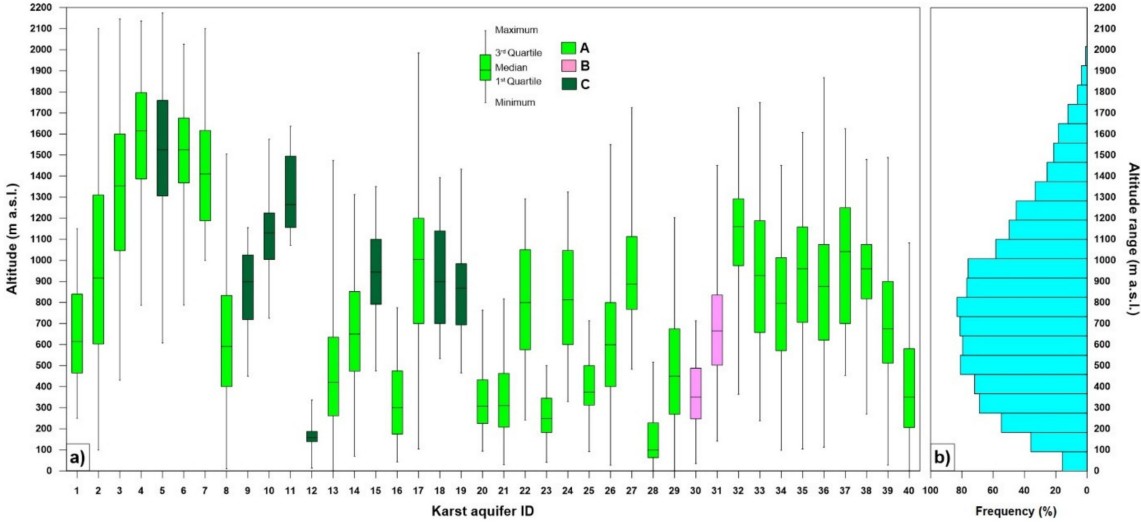

**Figure 2.** (**a**) Box plot of the altitude distributions of single karst aquifers (see the IDs in Table 1). Key to symbols: (**A**) limestone and dolomitic limestone units of the carbonate platform series (Jurassic–Paleogene); (**B**) dolomitic units of the carbonate platform series (Triassic–Liassic); (**C**) calcareous-marly units of the outer basin series (Triassic–Paleogene). (**b**) Frequency histogram of the altitude distribution of the 40 studied karst aquifers.

The high mean annual specific groundwater yield of karst aquifers of the southern Apennines, varying from 505 to 1104 mm·year$^{-1}$ (0.016 to 0.035 m$^3$ s$^{-1}$ km$^{-2}$) [32], is due to the high permeability caused by fracturing and karsts, as well as frequent endorheic morphologies that favor a very relevant groundwater recharge. The latter results in high values of the annual groundwater recharge coefficient (AGRC) (Table 1; [2]), which was estimated as the ratio between the mean annual net groundwater outflow (OUT = ($Qs + Qt$) + ($Uo + Ui$)) and the mean annual precipitation minus actual evapotranspiration ($P -$ AET), where both were related to the whole recharge area:

$$\text{AGRC} = \frac{Qs + Qt + Uo + Ui}{P - \text{AET}}, \tag{1}$$

where $Qs$ is the mean annual spring discharge, $Qt$ is the mean annual tapped discharge, $Uo$ is the mean annual groundwater outflow through adjoining aquifers, and $Ui$ is the mean annual groundwater inflow from adjoining aquifers and allogenic recharge.

For karst aquifers, which are characterized by a peculiar geomorphological feature, namely, by a summit plateau and endorheic areas with total infiltration and no runoff, an additional coefficient was assessed in order to estimate the recharge for the sloped areas only:

$$\text{AGRC}_s = \left[ \frac{(\text{AGRC} \times A_T) - (1 - A_E)}{A_T - A_E} \right] \times 100, \tag{2}$$

where $\text{AGRC}_S$ is the annual groundwater recharge coefficient for sloped areas, $A_T$ is the total area of the karst aquifer (km$^2$) and $A_E$ is the cumulative extension of the summit plateau areas and/or endorheic watersheds (km$^2$).

In addition, high values of mean annual specific groundwater yields are also due to climatic features, which are characterized by the Mediterranean-type climate with hot dry summers and moderately cool and rainy winters. Mean annual air temperatures range from approximately 10 to 12 °C in the mountainous interior to 13 to 15 °C in the coastal areas. The rainfall regimes vary from coastal to Mediterranean to the Apennine sublittoral

types [33], with the latter being characterized by a principal maximum in autumn–winter and a minimum in the summer. The distribution of precipitation over the region is mainly controlled by the Apennine mountain ridge, where orographic precipitation of humid air masses [34] coming eastwardly from the Tyrrhenian Sea dominates. Mean annual precipitations can reach up to 1700–2000 mm in the central part of the mountain chain. According to the Koppen–Geiger [35] classification, climate types vary across karst aquifers of southern Italy from warm-temperate (Csb) in the coastal areas to sub-continental temperate (Cfa) in the interior areas.

The climatic characteristics of southern Italy and their spatial and temporal variability strongly control the recharge processes in karst aquifers. Among the principal phenomena affecting the climatic variability across the study area, the North Atlantic Oscillation has been recognized as controlling the decadal variability of precipitation and groundwater recharge [5,6].

## 3. Data and Methodologies

### 3.1. Cartographic Database and the Precipitation and Air Temperature Time Series

This research was carried out in a large sector of the southern Apennines, covering approximately 19,339 km$^2$ (Figure 1). Based on preceding hydrogeological studies carried out for singular karst aquifers [2,30,36], 40 principal karst aquifers covering approximately 8560 km$^2$ were identified and characterized (Figure 1). In a GIS environment, the following datasets of the karst aquifers were implemented and analyzed, along with the time series of the annual AET: hydrogeological map of southern Italy, 1:250,000 scale [32]; digital elevation model (DEM) with a resolution of 20 × 20 m; Corine Land Cover Project [37]; land system map of the Campania Region, 1:250,000 scale [38]; annual normalized difference vegetation index (NDVI), which expresses the density of vegetation via the observation of distinct colors (wavelengths) of visible and near-infrared sunlight reflected by the plants [39].

Moreover, time series of the annual precipitation and air temperatures, from 2000 to 2014, recorded by the Protezione Civile meteorological network (266 rain gauge stations, 150 air temperature stations, and 150 thermo-pluviometric stations) were considered (Figure 3).

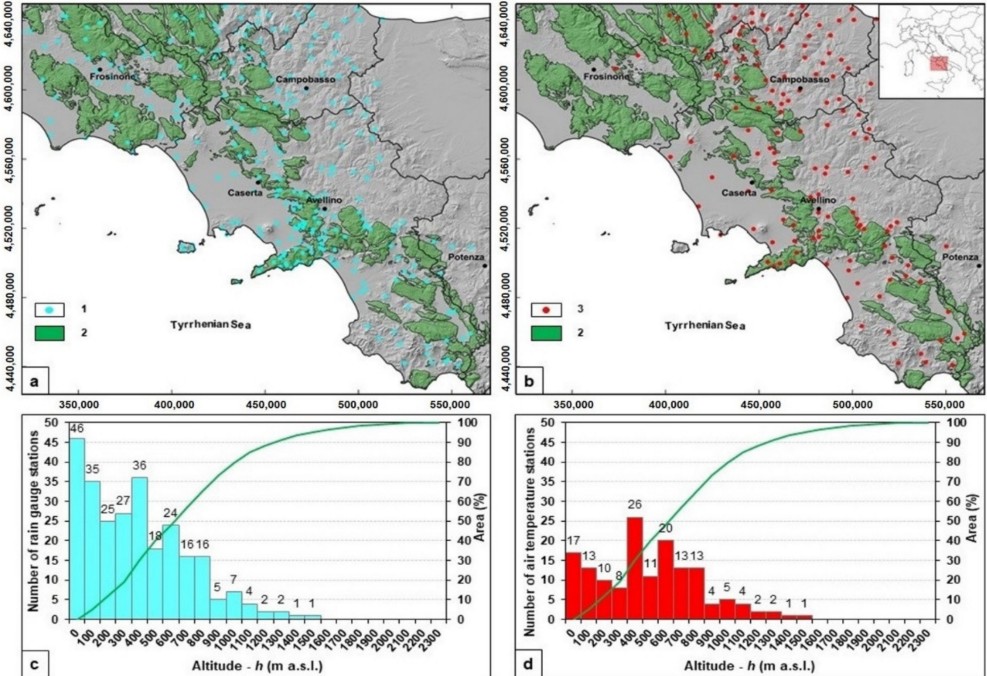

**Figure 3.** Spatial distributions of rain gauges (**a**,**c**) and air temperature stations (**b**,**d**).

By means of the regression kriging method [40], mean annual rainfall and air temperature regional distributed models were reconstructed and implemented (Figure 4), thereby accounting for variations due to the orographic control of mountain ranges [41] and altitude [42].

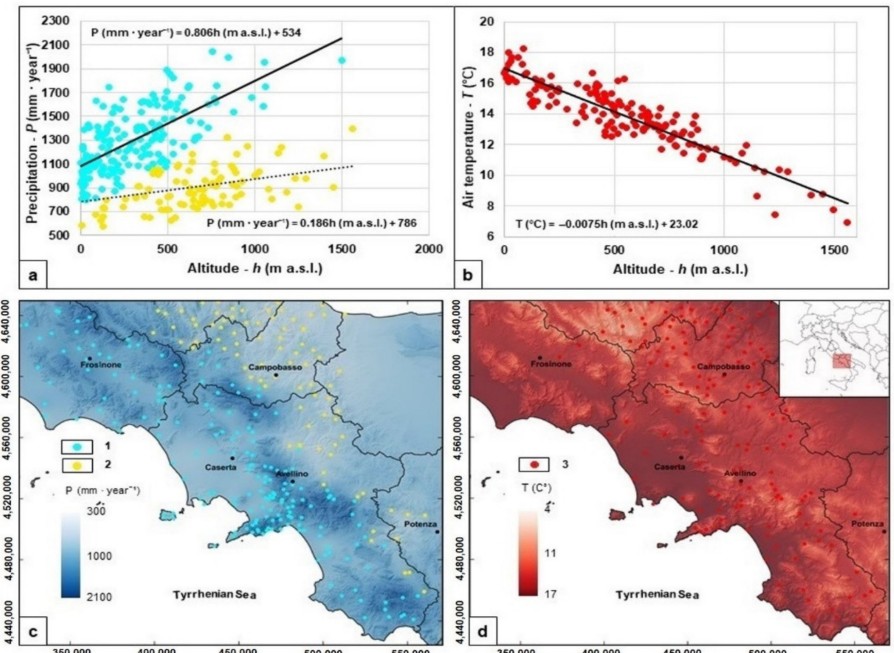

**Figure 4.** (**a**) Regression model of the mean annual precipitation (2000–2014) with altitude (blue points represent rain gauge stations included in the windward pluviometric zone, while yellow points represent rain gauge stations in the leeward pluviometric zone); (**b**) regression model of the mean annual air temperature (2000–2014) with altitude; (**c**) distributed model of the mean annual rainfall; (**d**) distributed model of the mean annual air temperature.

### 3.2. Estimation of Evapotranspiration Using Remotely Sensed Data and Classical Empirical Formulas

Among the variety of data and satellite platforms available, the MODIS AET datasets (annual MOD16A3) obtained by the University of Montana's Numerical Terradynamic Simulation Group (ftp.ntsg.umt.edu/pub/MODIS) were considered suitable for the scope of this research, especially for the spatial resolution (1000 × 1000 m), which was considered appropriate for the extent of a regional study area.

These datasets were produced using an improved algorithm that was applied to remote sensing and meteorological data, which allows for the estimation of the daily AET [43].

The original algorithm, named MOD16 ET [44], is based on the Penman-Monteith equation [45]:

$$\lambda E = \frac{sA + \rho C_p \frac{e_{sat} - e}{r_a}}{s + \gamma \left(1 + \frac{r_s}{r_a}\right)} \tag{3}$$

$\lambda E$ (W m$^{-2}$) is the latent flux, where $\lambda$ (J kg$^{-1}$) is the latent heat of evaporation; $s$ (Pa K$^{-1}$) = $d(e_{sat})/dT$ is the slope of the curve relating the saturated water vapor pressure ($e_{sat}$ (Pa)) to temperature (K); $A$ (W m$^{-2}$) is the available energy; $\rho$ (kg m$^{-3}$) is the air density; $C_p$ (J kg$^{-1}$ K$^{-1}$) is the specific heat capacity of the air; $e$ (Pa) is the actual water vapor pressure; $r_a$ (s m$^{-1}$) is the aerodynamic resistance; $\gamma$ (Pa K$^{-1}$) = $(M_a/M_w)(C_p P/\lambda)$, where $M_a$ (kg mol$^{-1}$) and $M_v$ (kg mol$^{-1}$) are the molecular masses of dry air and wet air, respectively, and $P$ (Pa) is the atmospheric pressure; $r_s$ (s m$^{-1}$) is the surface resistance which

is an effective resistance to evaporation from the soil surface and transpiration from the plant canopy.

Specifically, the algorithm considers the effects of both the surface energy partitioning process and the environmental controls on evapotranspiration. The algorithm estimates the AET using ground-based meteorological observations and MODIS data and by (1) adding the vapor pressure deficit and minimum air temperature constraints onto the stomatal conductance, (2) using the leaf area index (LAI) as a scalar for estimating canopy conductance, (3) replacing the NDVI with the enhanced vegetation index (EVI) by also changing the equation for calculating the vegetation cover fraction (FC), and (4) considering soil evaporation using a model based on MODIS data [46] whose reliability was successfully tested via a comparison with measurements of two flux towers in Australia. The MODIS16 ET algorithm was validated by observations of evapotranspiration using 19 AmeriFlux eddy covariance flux towers.

The former form of the algorithm (MODIS16 ET) calculated the AET as the sum of the evaporation from moist soil and the transpiration from the vegetation during the daytime. In the new form, the algorithm also considers the AET during the nighttime, as well as the vegetation cover fraction, the stomatal conductance, the aerodynamic conductance, etc. [43].

To test the improvements of the application of the MODIS AET when assessing the groundwater recharge of karst aquifers of southern Italy, we compared it to values of the mean annual AET calculated using classical approaches based on the empirical Coutagne formula (1954; Equation (4)) [47]:

$$AET_{ji} = AP_{ji} - \frac{1}{0.8 + 0.14AT_{ji}}, \tag{4}$$

and Turc formula (1954; Equation (5)) [48]:

$$AET_{ji} = \frac{AP_{ji}}{\sqrt{0.9 + \left(\frac{AP_{ji}}{300 + 25 \times AT_{ji} + 0.05 \times AT_{ji}^3}\right)^2}}, \tag{5}$$

where:

$ETR_{ji}$—real evapotranspiration for the jth rain gauge station and the ith year (mm).

$AP_{ji}$—annual precipitation for the jth rain gauge station and the ith year (mm).

$AT_{ji}$—annual air temperature for the jth rain gauge station and the ith year ($°C$).

Moreover, we compared MODIS AET with the PET estimated using the Thornthwaite formula (1955; Equation (6)) [49]:

$$PET_{ji} = K \times 16 \times \left(\frac{10T_{mji}}{I}\right)^\alpha, \tag{6}$$

where:

$PET_{ji}$—potential evapotranspiration for the jth rain gauge station and the ith month (mm).

$K$—coefficient that depends on the monthly average of hours of insolation and a function of the latitude and month.

$T_{mji}$—mean monthly air temperature ($°C$).

$I = \left(\frac{T_{mji}}{5}\right)$—annual thermal index that is given by the sum of the monthly thermal indices, where each is expressed by:

$$\alpha = 675 \times 10^{-9} \times I^3 - 771 \times 10^{-7} \times I^2 + 1792 \times 10^{-5} \times I + 0.49239. \tag{7}$$

These formulas were applied to the annual data that was recorded by rain gauge and air temperature stations located across the whole recharge area and then averaged on a yearly basis.

### 3.3. Groundwater Recharge Estimation

The mean annual groundwater recharge of the 40 karst aquifers of southern Italy was estimated based on the reconstruction of distributed models of precipitation and evapotranspiration, with the latter being related to the application of the Coutagne, Turc, and Thornthwaite formulas and estimates derived from the MODIS data. Considering the difference between precipitation and evapotranspiration models, distributed models of effective precipitation (*P*-ET) were reconstructed. Subsequently, considering the AGRC values estimated for the karst aquifers of the southern Apennines (Table 1; [2]), four distributed groundwater recharge models were reconstructed at a mean annual scale (2000–2014) by taking into account the respective models of evapotranspiration that are derived using Coutagne, Turc, and Thornthwaite formulas, as well as MODIS data.

## 4. Results

### 4.1. Distributed Modelling of Precipitation and Air Temperature

To generate distributed models of precipitation and air temperature across the study area, while accounting for the inhomogeneous planimetric and altimetric distribution of rain gauges and air temperature stations over the territory, regression models with altitude were carried out. This analysis was conceived as being based on the consistent correlation of both variables with altitude, as well as being useful for recognizing different pluviometric zones, depending on the effects of the orographic and physiographic features [34].

Regarding precipitation, the correlation with altitude (Figure 4a) and prevalent eastward movement of humid air masses coming from the Atlantic Ocean allowed for recognizing two pluviometric zones. A windward pluviometric zone, characterized by higher precipitation, extended from the Thyrrhenian Sea coastline to the principal morphological divide of the Apennine chain. A leeward pluviometric zone, characterized by lower precipitation, included the area eastward of the principal morphological divide. For both pluviometric zones, a linear regression model with altitude was found, even if it was characterized by a relevant difference due to the orographic effect, or rain shadow effect, due to the Apennine chain.

For the windward pluviometric zone:

$$P \text{ (mm)} = 0.806 \times h \text{ (m a.s.l.)} + 534 \quad (\text{corr.} = 0.707; \text{prob.}_{\text{t-Student}} < 0.001\%) \tag{8}$$

For the leeward pluviometric zone:

$$P \text{ (mm)} = 0.187 \times h \text{ (m a.s.l.)} + 786 \quad (\text{corr.} = 0.289; \text{prob.}_{\text{t-Student}} = 0.36\%) \tag{9}$$

Considering the extent of the windward pluviometric zone, which included the 40 principal karst aquifers of the study area, the related linear correlation between precipitation (*P*) and altitude (*h*) (Equation (8)) was considered suitable for reconstructing the distributed model of precipitation across the study area using a regression kriging technique (Figure 4c).

Different from the precipitation, air temperature showed a unique linear correlation with the altitude, while not being influenced by the slope aspect or the direction of movement of humid air masses (Figure 4b):

$$T \text{ (°C)} = -0.0075 \times h \text{ (m a.s.l.)} + 23.02 \quad (\text{corr.} = -0.914; \text{prob.}_{\text{t-Student}} < 0.001\%). \tag{10}$$

This empirical linear relationship was implemented in a regression kriging method to obtain a distributed model of air temperature across the study area (Figure 4d).

From both distributed models, the mean values of precipitation and air temperature were estimated for the areas of the karst aquifers with the same spatial resolution (1000 × 1000 m) in the MODIS AET data.

### 4.2. Distributed Models of the Mean Annual AET

To estimate the groundwater recharge at the regional scale, distributed models of the mean annual (2000–2014) AET, calculated using Coutagne and Turc formulas, as well as the mean annual PET, calculated with the Thornthwaite formula, were reconstructed based on distributed models of precipitation and air temperature. Moreover, a distributed model of the mean annual MODIS AET was also reconstructed for the same period. The spatial resolution of the Coutagne, Turc, and Thornthwaite models was homogenized with that of the MODIS AET (1000 × 1000 m). To assess the spatial variability of the mean annual AET, the values calculated for each pixel were statistically analyzed using aggregations for both single and all karst aquifers.

Among the initial results is the estimation of the mean annual values of AET for the aggregated areas of the 40 karst aquifers considered. In particular, the MODIS AET was found to correspond to about 670 mm·year$^{-1}$, while the Coutagne, Turc, and Thornthwaite formulas were found to correspond to about 599 mm·year$^{-1}$, 539 mm·year$^{-1}$, and 694 mm·year$^{-1}$, respectively.

Moreover, the spatial variability of the mean annual MODIS AET was estimated using box plots for single karst aquifers and using a frequency analysis of all karst aquifers aggregated (Figure 5).

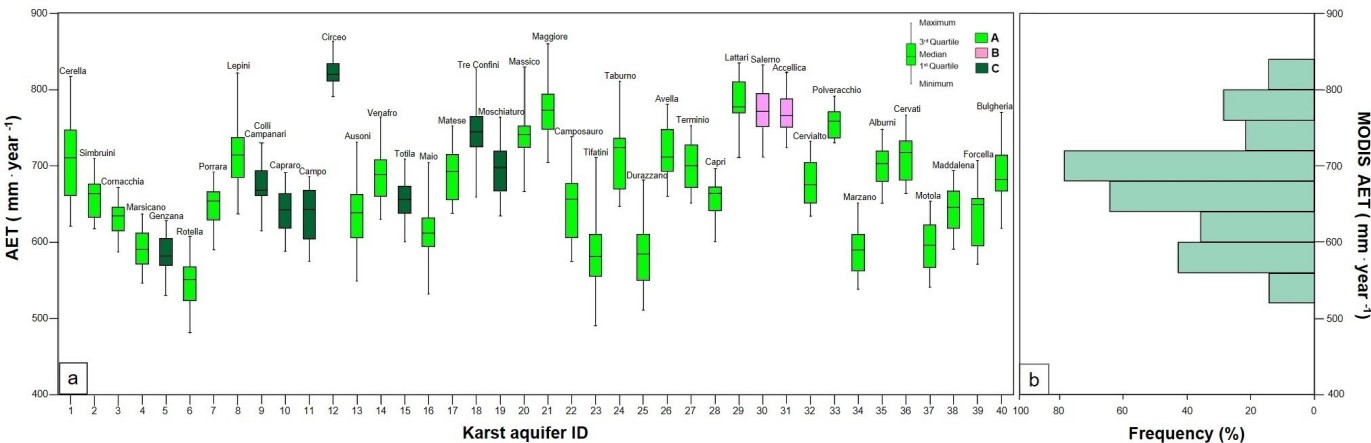

**Figure 5.** (**a**) Box plot of the Moderate Resolution Imaging Spectrometer (MODIS) actual evapotranspiration (AET) [43] estimates for each karst aquifer. Keys to legend: limestone and dolomitic limestone units of the carbonate platform series (Jurassic–Paleogene); dolomitic units of the carbonate platform series (Triassic–Liassic); calcareous-marly units of the outer basin series (Triassic–Paleogene). (**b**) Frequency histogram of the MODIS AET for all 40 karst aquifers aggregated.

Via an analysis of the spatial variability of the MODIS AET for the 40 karst aquifers studied, the Mt. Circeo karst aquifer (ID 12) was recognized as having the highest value of mean annual MODIS AET (820 mm·year$^{-1}$), while the minimum value of the mean annual MODIS AET (about 550 mm·year$^{-1}$) was recognized for the Mt. Rotella karst aquifer (ID 6).

The relevant spatial variability of the MODIS AET among the 40 karst aquifers considered (Figure 5) was presumed to be related to the variability of the parameters controlling the evapotranspiration, such as precipitation and air temperature, which are dependent on the altitude and land use or vegetation type (accounting for soil moisture availability). In this regard, the Pearson correlations between the mean annual value of the MODIS AET, estimated for each karst aquifer, and other mean parameters estimated for each karst aquifer, such as air temperature, precipitation, NDVI, altitude, forest coverage, distance

from the coastline, and percentage of outcropping limestone lithology, was carried out (Figure 6).

**Figure 6.** Correlation matrix among the MODIS AET and other hydrological and physiographic variables. NDVI—normalized difference vegetation index.

The analysis of the correlation matrix revealed complex relationships between the parameters that depend on the altitude, thus on the morphological and physiographic settings of the study area, as well as the vegetation cover. The following parameters were best correlated with the MODIS AET and were of higher statistical significance: NDVI (corr. = 0.83), which directly controls evapotranspiration demand; altitude (corr. = −0.45), which indirectly regulates air temperature; air temperature (corr. = 0.44), which directly controls the evapotranspiration process; forest coverage (corr. = 0.43), which directly regulates the vegetation density and NDVI; distance from the coast line (corr. = −0.45), which indirectly controls the altitude because it depends on the physiographic setting of the region studied; precipitation (corr. = 0.36), which regulates the availability of the soil water for the evapotranspiration demand. In contrast, the correlation with the percentage of outcropping limestone lithology was found to be an insignificant statistical correlation due to the mixed composition of carbonate mountains, which were also formed by dolomite rocks.

The mean annual AET values, which were estimated using the MODIS algorithm [43], Coutagne, and Turc formulas, and the mean annual PET values, which were estimated using the Thornthwaite formula, were mutually compared for each karst aquifer (Figure 7); this allowed for assessing, in general, the scatter of the values obtained with the empirical formulas to see whether they were consistent with those from the MODIS AET (Figure 5). Moreover, the estimations carried out using the Coutagne and Turc formulas tended to give lower values, while those obtained using the Thornthwaite formula and the MODIS AET produced higher values.

To better compare the results of the AET obtained using different methods, pairwise frequency analysis of the differences between the estimates of the mean annual MODIS AET, the AET estimated using the Turc and Coutagne formulas, and the PET using the Thornthwaite formula were calculated (Figure 8). Considering the mean value of the pairwise difference as the parameter indicating the best match, the results obtained using the Thornthwaite formula were found to be the nearest to those of the MODIS AET, as characterized by the lowest value (−19.0 mm). In contrast, the results obtained using

the Turc and Coutagne formulas were recognized as being characterized by higher mean differences, respectively 76.5 mm and 137 mm. Moreover, the frequency analysis of pairwise differences showed very similar Gaussian distributions with the standard deviation values ranging narrowly between 140 and 183 mm.

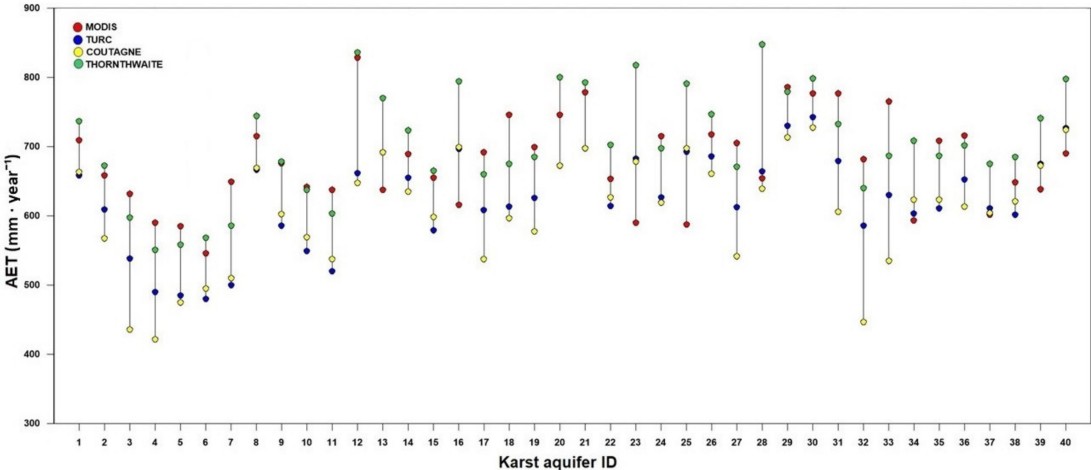

**Figure 7.** Comparison of the AET estimates using the Coutagne and Turc formulas, potential evapotranspiration (PET) using the Thornthwaite formula, and the MODIS AET.

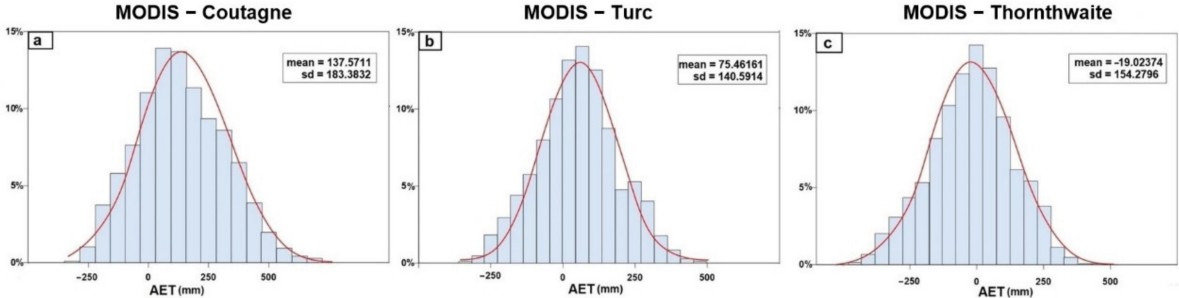

**Figure 8.** Frequency analysis of the pairwise differences between estimates of the mean annual MODIS AET and values of the AET calculated using the Coutagne formula (**a**), Turc formula (**b**), and the values of PET estimated using the Thornthwaite formula (**c**).

### 4.3. Groundwater Recharge Assessment

Different estimations of the groundwater recharge for the 40 karst aquifers of the study area were obtained by considering the distributed model of precipitation (Figure 4c); distributed models of evapotranspiration, as estimated using the Coutagne, Turc, and Thornthwaite formulas; distributed model of the MODIS AET and AGRC values of the karst aquifers (Table 1). As result, four respective distributed models of groundwater recharge were calculated, which were based on the evapotranspiration estimates found using the Coutagne, Turc, and Thornthwaite formulas, as well as the MODIS satellite data (Figure 9).

As a preliminary result, by considering the aggregation of all karst aquifers, the estimate of the mean annual groundwater recharge based on the MODIS AET was 448 mm·year$^{-1}$. In contrast, by considering the Coutagne, Turc, and Thornthwaite formulas, it was estimated as being 533, 494, and 437 mm·year$^{-1}$, respectively.

Furthermore, for the groundwater recharge, the estimations were mutually compared for each aquifer (Figure 10; Table 2). From this comparison, the relevant scatter of the results was recognized as being principally controlled by the precipitation, namely, by the

altitude and physiographic factors, and secondarily by the AET (Figure 7) and AGRC; thus, the variability was conceptually consistent with that of parameters previously illustrated.

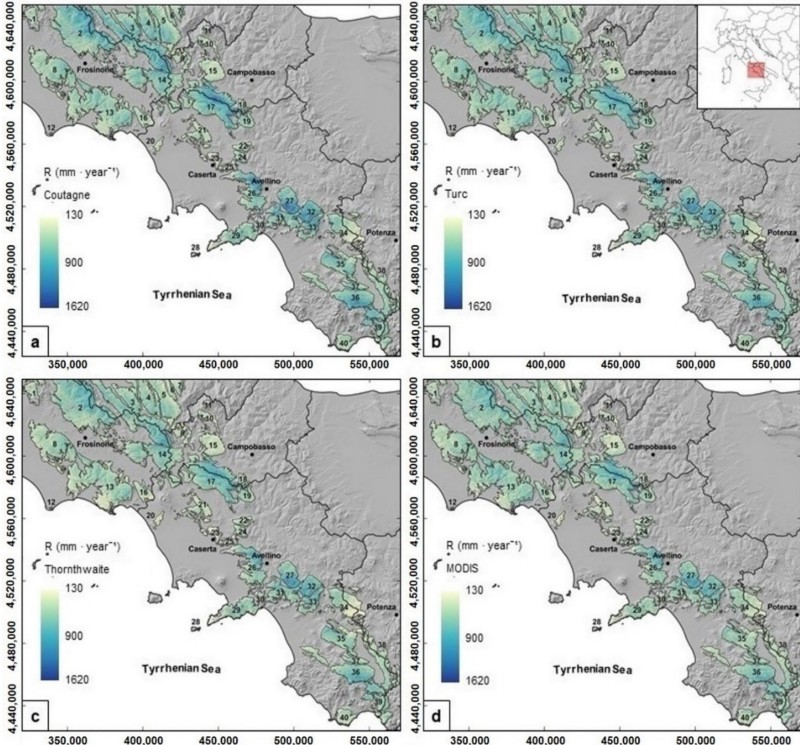

**Figure 9.** Distributed models of the groundwater recharge of karst aquifers that were estimated by considering the evapotranspiration calculated using the (**a**) Coutagne formula, (**b**) Turc formula, (**c**) Thornthwaite formula, and (**d**) MODIS satellite data.

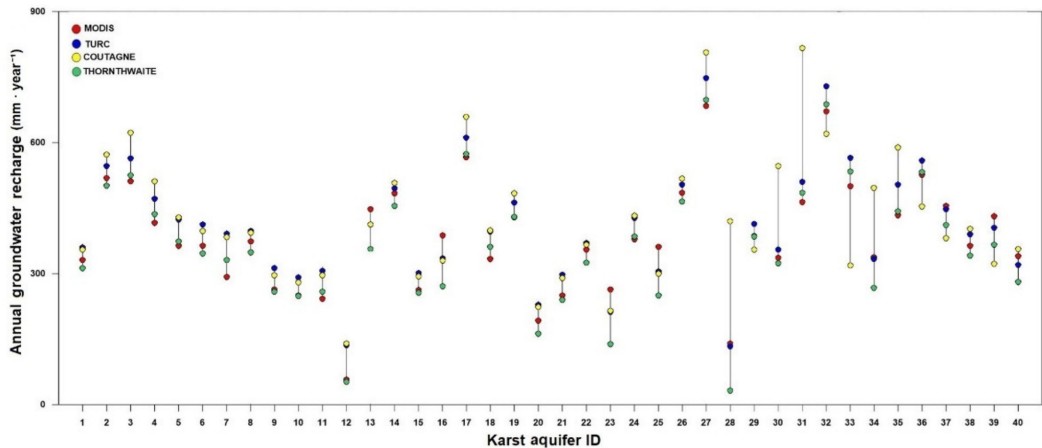

**Figure 10.** Comparison of the mean annual groundwater recharge that was estimated by considering the MODIS AET, as well as the Coutagne, Turc, and Thornthwaite formulas.

Regarding the effects of considering different approaches for the estimation of the AET, the highest values of the mean annual groundwater recharge were obtained for the Mt. Terminio karst aquifer (ID 27), while the lowest values were found for the Mount Circeo karst aquifer (ID 12). This difference could be explained by the difference in altitude of two aquifers, thus by the different mean annual precipitation, as well as by the relevantly higher AGRC value for the Terminio karst aquifer (78%), enhanced by the relevant occurrence of endorheic areas, in comparison to that of the Circeo karst aquifer (48%). Moreover, after analyzing the results obtained for each karst aquifer, the values of groundwater

recharge obtained using the Coutagne and Turc formulas were generally found to be the highest ones, not considering very few exceptions (IDs 13, 16, 39, and 40). Instead, estimates that used the Thornthwaite formula and MODIS satellite data were recognized as the lowest ones. This difference was consistent with that observed for the estimates of evapotranspiration carried out with the approach considered.

**Table 2.** Values of the mean annual groundwater recharge ($10^6$ m$^3$·year$^{-1}$) of the karst aquifers of the study area (Figure 1) as calculated using different estimates of evapotranspiration: AET (Coutagne and Turc formulas), PET (Thornthwaite formula), and MODIS AET.

| ID | Karst Aquifer | Area (km$^2$) | Coutagne ($10^6$ m$^3$·year$^{-1}$) | Turc ($10^6$ m$^3$·year$^{-1}$) | Thornthwaite ($10^6$ m$^3$·year$^{-1}$) | MODIS AET ($10^6$ m$^3$·year$^{-1}$) |
|----|---------------|---------------|-------------------------------------|---------------------------------|------------------------------------------|--------------------------------------|
| 1 | Cerella | 137 | 51.7 | 51.3 | 42.6 | 44.3 |
| 2 | Simbruini | 1076 | 664.9 | 611.7 | 554.3 | 561.2 |
| 3 | Cornacchia | 723 | 464.5 | 410.9 | 380.7 | 366.9 |
| 4 | Marsicano | 204 | 111.5 | 96.6 | 88.6 | 84.0 |
| 5 | Genzana | 277 | 128.7 | 116.8 | 102.2 | 98.2 |
| 6 | Rotella | 40 | 17.6 | 16.6 | 13.9 | 14.6 |
| 7 | Porrara | 64 | 26.1 | 25.0 | 21.1 | 18.3 |
| 8 | Lepini | 483 | 211.1 | 206.0 | 175.2 | 180.7 |
| 9 | Colli Campanari | 97 | 30.9 | 30.8 | 25.0 | 25.4 |
| 10 | Capraro | 61 | 19.4 | 19.4 | 14.8 | 14.7 |
| 11 | Campo | 16 | 5.1 | 5.0 | 4.2 | 3.8 |
| 12 | Circeo | 7 | 1.0 | 1.0 | 0.4 | 0.5 |
| 13 | Ausoni | 826 | 374.8 | 368.1 | 302.3 | 372.4 |
| 14 | Venafro | 365 | 193.8 | 189.8 | 168.3 | 174.9 |
| 15 | Totila | 195 | 59.2 | 59.4 | 49.8 | 50.4 |
| 16 | Maio | 93 | 33.4 | 32.8 | 25.0 | 35.3 |
| 17 | Matese | 588 | 412.7 | 367.3 | 342.5 | 331.2 |
| 18 | Tre Confini | 28 | 11.5 | 11.0 | 9.9 | 8.8 |
| 19 | Moschiaturo | 85 | 41.9 | 39.9 | 36.2 | 35.8 |
| 20 | Massico | 29 | 7.1 | 7.0 | 4.5 | 5.6 |
| 21 | Maggiore | 173 | 53.9 | 52.8 | 41.8 | 42.9 |
| 22 | Camposauro | 50 | 19.8 | 19.2 | 16.1 | 17.7 |
| 23 | Tifatini | 65 | 14.8 | 14.4 | 8.6 | 17.0 |
| 24 | Taburno | 43 | 20.4 | 19.3 | 17.2 | 16.4 |
| 25 | Durazzano | 52 | 17.1 | 16.8 | 13.4 | 19.1 |
| 26 | Avella | 334 | 212.5 | 188.3 | 172.1 | 174.3 |
| 27 | Terminio | 167 | 143.3 | 127.9 | 117.7 | 114.0 |
| 28 | Capri | 9 | 1.4 | 1.3 | 0.4 | 1.2 |
| 29 | Lattari | 245 | 118.6 | 110.7 | 99.9 | 98.2 |
| 30 | Salerno | 46 | 18.8 | 17.8 | 16.1 | 16.2 |
| 31 | Accellica | 206 | 121.9 | 109.4 | 102.2 | 96.9 |
| 32 | Cervialto | 129 | 110.8 | 94.3 | 89.0 | 85.6 |
| 33 | Polveracchio | 114 | 75.9 | 65.7 | 61.5 | 56.4 |
| 34 | Marzano | 308 | 108.1 | 105.2 | 82.7 | 103.1 |
| 35 | Alburni | 254 | 139.8 | 133.2 | 113.0 | 109.1 |
| 36 | Cervati | 318 | 197.1 | 181.8 | 163.3 | 160.8 |
| 37 | Motola | 52 | 25.6 | 23.9 | 20.9 | 23.1 |
| 38 | Maddalena | 300 | 124.7 | 122.8 | 102.9 | 109.1 |
| 39 | Forcella | 217 | 97.0 | 94.4 | 80.3 | 92.4 |
| 40 | Bulgheria | 101 | 38.4 | 37.0 | 29.8 | 35.1 |

In order to assess the effect of considering the evapotranspiration calculated using the Coutagne, Turc, and Thornthwaite formulas on the estimation of the groundwater recharge, a pairwise frequency analysis of differences with estimates based on the MODIS AET was performed (Figure 11). According to the comparison of the results for the estimates of the evapotranspiration, the mean value of the pairwise differences with estimates based on

the MODIS AET was considered to indicate the performances of the classical formulas of Coutagne, Turc, and Thornthwaite well.

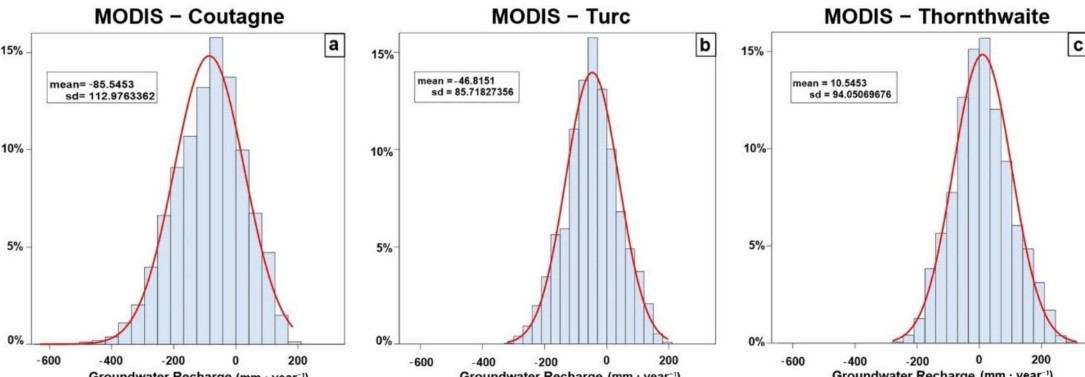

**Figure 11.** Statistical distribution of the pairwise differences in the estimates of the mean annual groundwater recharge calculated by considering the evapotranspiration using the Coutagne (**a**), Turc (**b**), and Thornthwaite (**c**) formulas in comparison to that based on the MODIS AET.

Furthermore, in the case of the estimations of groundwater recharge, the lowest values of the mean difference were found for estimates based on the Thornthwaite formula, corresponding to 10.5 mm·year$^{-1}$. In contrast, higher values of the mean difference were found for estimates based on the Turc and Coutagne formulas, with $-46.8$ mm·year$^{-1}$ and $-85.5$ mm·year$^{-1}$, respectively.

The frequency analysis of the pairwise differences showed a very similar Gaussian distribution with standard deviations varying from 85.7 to 112.9 mm·year$^{-1}$.

## 5. Discussion

Several examples of groundwater recharge assessments by means of remote sensing data are found in the literature for different parts of the world [50–54], as well as for identifying sites with the most effective artificial groundwater recharge [55]. The data and approaches used in these studies were principally addressed at estimating the hydrological parameters related to soil water availability and evapotranspiration and at processing these data in a GIS environment. None of these cases involved the study of karst aquifers; therefore, the approaches and results shown in this research can be considered significant in the field of karst hydrogeology.

Given this recognition of the scientific literature, the results obtained via the modeling of parameters that control groundwater recharge in karst aquifers of southern Apennines revealed useful advances in the management of these types of estimates, which are of fundamental relevance in the framework of an appropriate assessment of these groundwater resources.

A first important result achieved in reconstructing the regional distributed models of groundwater recharge was the recognition of the orographic barrier effect of the Apennine chain, which strongly controls the spatial distribution of the mean annual precipitation, via the identification of windward and leeward pluviometric zones, where each zone provided a specific correlation of precipitation with altitude. This outcome, coupled with the unique linear correlation found between the mean annual air temperature and altitude, was proposed as a useful tool for modeling spatially distributed precipitation and air temperature via a regression kriging technique. These distributed models were used to estimate evapotranspiration via the classical formulas of Coutagne, Turc, and Thornthwaite.

The interannual variabilities of precipitation and air temperature were found to be the main parameters controlling the AET at the aquifer scale and across the whole study area. Accordingly, the results of the MODIS AET were found to be characterized by strong spatial and temporal variability when considering both the annual and mean annual time scales.

The results obtained regarding the estimation of groundwater recharge of the 40 karst aquifers of southern Italy using the integration of the AET estimated via MODIS satellite data opens new perspectives and advances in the assessment of groundwater resources. The most important result is the overcoming of uncertainties related to the application of empirical formulas due to spatially and temporally discontinuous meteorological data. In particular, the comparison of the results obtained when considering the MODIS AET with those derived using the application of classical formulas indicated that the MODIS AET was higher than that estimated using the Coutagne and Turc empirical methods, while it was very close to the values of PET estimated using the Thornthwaite formula.

This result gives very interesting hints and understanding regarding the close correspondence of the AET to the PET, which appears to be dependent on (a) relevant precipitation across the mountain ranges forming the karst aquifers in southern Italy; (b) limited evapotranspiration demand due to the lower temperature, which is due to the higher altitudes; (c) the existence of a diffused and quite continuous soil covering across the karst areas, which favors the development of dense vegetation that is often characterized by forest land use. An important contribution to the formation of the soil cover in the study area was related to the activity of volcanic centers of the Campania region during Quaternary, whose explosive eruptions led to the dispersion of ash-fall pyroclastic deposits with a spatial variable distribution across the study area that depended on the dispersal axes and distance from volcanic vents.

Under these conditions, soils covering the karst aquifers behave as a water tank, which stores moisture during the autumn–winter rainy season and releases it during the spring–summer season for the evapotranspiration demand. The results obtained indicate that, in general, the evapotranspiration demand, or PET, was satisfied by the availability of water resources from soil water storage and was almost equivalent to the AET.

## 6. Conclusions

Through the comparison with the classical approaches used for the estimation of evapotranspiration, the application of the MODIS satellite data was demonstrated to be a practical tool to estimate the AET and reduce the uncertainty due to the spatial and temporal inhomogeneity of meteorological networks. Therefore, the results achieved allowed for obtaining advances in the assessment of the groundwater recharge of the karst aquifers at a regional scale and to understand how climatic conditions and soil and vegetation features existing across karst aquifers of southern Italy allow for the actual evapotranspiration to be close to corresponding with the potential evapotranspiration.

The integration of the hydrological terrestrial data with those derived from the MODIS satellite represents a valid approach for the estimation and modeling of AET values, and therefore, the groundwater recharge at regional and mean annual scales. This finding would be very helpful for upscaling the results of further studies to be carried out on the analysis of hydrological processes occurring at the local scale in different conditions of soil covering and bedrock fracturing/karstification.

Finally, this method can be conceived as an important tool for an appropriate management model of groundwater of karst aquifers that is aimed at controlling and mitigating the effects of climate variability and can be used for other karst aquifers around the world that are characterized by the inconsistent availability of meteorological recordings.

**Author Contributions:** Conceptualization, G.R., P.D.V., V.A., F.B., D.C. and P.M.; methodology, G.R., P.D.V., V.A., F.B., D.C. and P.M.; software, G.R., P.D.V., V.A., F.B., D.C. and P.M. and validation, G.R., P.D.V., V.A., D.C. and P.M., G.R., P.D.V., V.A., F.B., D.C. and P.M.; formal analysis, G.R., P.D.V., V.A., F.B., D.C. and P.M.; investigation, G.R., P.D.V., V.A., F.B., D.C. and P.M.; funding acquisition, P.D.V., V.A.; data curation, G.R., P.D.V., V.A., F.B., D.C. and P.M.; writing—original draft preparation, G.R., P.D.V., V.A., F.B., D.C. and P.M. and writing—review and editing, G.R., P.D.V., V.A., F.B., D.C. and P.M.; visualization, G.R., P.D.V., V.A., F.B., D.C. and P.M.; supervision, G.R., P.D.V., V.A., F.B., D.C. and P.M.; project administration, G.R., P.D.V., V.A., F.B., D.C. and P.M. All authors have read and agreed to the published version of the manuscript.

**Funding:** This research was funded by the Earth Science, Environment and Resources, PhD Program of the University of Naples Federico II (XXX cycle).

**Institutional Review Board Statement:** Not applicable.

**Informed Consent Statement:** Not applicable.

**Data Availability Statement:** Not applicable.

**Acknowledgments:** The authors are grateful to the Department of Civil Protection of the Abruzzo, Molise, Campania, Basilicata, and Apulia regions (Italy), which kindly provided the rainfall and temperature data. We are also thankful to the University of Montana's, Numerical Terradynamic Simulation Group for the MODIS data provided.

**Conflicts of Interest:** The authors declare no conflict of interest.

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
