# Peer review of "Testing Evapotranspiration Estimates Based on MODIS Satellite Data in the Assessment of the Groundwater Recharge of Karst Aquifers in Southern Italy"

_water, doi:10.3390/w13020118_

Round 1

Reviewer 1 Report

The manuscript deals with recharge to carbonate aquifers, which dominate in many worlds’ áreas and especially in the Mediterranean areas, both in the north and in the south. The area considered in central-southern Italy is comparatively humid, with good soil and vegetation cover. This is a main difference with respect other drier areas, which are not dealt with.

The main objective is to evaluate the potential of MODIS satellite sensors to determine the spatial and time variability of recharge. Here recharge is considered to be the in-transit recharge, which leaving the soil and that later on will become aquifer recharge. This difference is not essential in the study as recharge values are aggregated annually.

190-192 the formula is dimensionally incongruent and the symbols differ from those in the explanation

The number of precipitation and temperature stations are given. How many of them are complete ones to apply the P-M ETP calculation?

251-258  the procedure should be presented more in detail for readers that are not experts in the matter. Potential evapotranspiration is shortened by ETP and real (actual) evapotranspiration by ETR, which is acceptable and I prefer, but it is worthwhile to indicate that many other authors use PET and AET, respectively

264-268   to avoid mistakes, indicate clearly which formula is each one

330  Fig.6 At a first glance the four maps seem equal. Thus they do not add anything relevant and can be deleted. What follows explains enough the differences

342   Fig.7 Indicate in the caption A,B,C and the colors

373  It is not surprising that Thornthwaite ET is the greatest as it is a potential value and does not intend to be a real (actual) value. This does not help to evaluate the correctness of the MODIS ETR values

375  Is Fig9 needed? It seems more appropriate to give and compare the statistical values

387  Figure 10 provides the same as figure 9 and more clearly

399  Fig.11 At a first glance the four maps seem equal. Thus they do not add anything relevant and can be deleted. Figure 13 (432) explains enough the differences. Is figure 12 needed?

The correctness of MODIS in-transit recharge should be compares with soil water balances results, if possible with complete weather stations, and applying Thornthwaite ETP calculations. This needs considering soil characteristics besides soil humidity and vegetation. The difference when there are ash-fall pyroclasts is mentioned twice in the manuscript. In the considered area, as it is relatively humid, it is possible that ETR is only slightly less than ETP, but this is not discussed. Consider that often Thornthwaite ETP tends to be underestimated in conditions similar to the area considered in the manuscript.

The manuscript will improve by considering the uncertainty of the recharge valus, as done in other papers.

SOME MINOR ERRORS AND COMMENTS

77  review the sentence

92  review the sentence

93  atmosphere

164  surrounding

177 m2s-1km-2  It will help giving the values also in mm/yr

185  -Ui

202  delete (1954)

217  DNVI is still undefined here; this is done in 252; not all readers will know what it means

387  Frequency

Reviewer 2 Report

Very interesting analyzes and conclusions that undoubtedly deserve publication. The combination of traditional research methods with remote sensing is the right direction in research on evapotranspiration and groundwater recharge. As shown by the authors, this analytical approach works well for large areas.

The article has a logical structure. The justification for the presented analysis is exhaustive. The characteristics of the research area have been presented synhetically and sufficiently. The physico-geographical elements that affect the water cycle are described.

Figures, maps and table are legible. The literature contains well-chosen citations. There are both general citations related to the researched issues and regional studies from the studied area. Also included are the earlier works of the co-authors Allocca V. and De Vita P. (they also appear as co-authors in 11 quoted articles).

Below are some minor corrections that should be taken into account before going to press:

The databases must be updated. Land use and weather conditions have intensified and strongly influenced by humans in recent years.

  • CORINE Land Cover from 2006 is out of date. A year later, the 2012 and 2018 editions were available. The current map from 2018 should be used in the study
  • Temperature and precipitation are analyzed from 2000 to 2014. Data from the last few years should be taken into account. These are very important data in the context of global climate change.

Supplementing the conclusions

  • In conclusion, it is also worth writing clearly in what direction further research on the Assessment of Groundwater Recharge of Karst Aquifers of Southern Italy should go?
  • It is also worth writing whether the presented research has a chance to be successfully applied in other parts of the world with similar geological characteristics?

Reviewer 3 Report

The article deals with interesting analysis of actual evapotranspiration  and recharge of karst aquifers of Southern Italy. A sufficient monitoring data for the period of 2000-2014 have been considered and overall the results are supported enough by satellite and hydrological observations. ET evaluation is always extremely important in hydrological / hydrogeological calculations and has been greatly developed in recent years thanks to remote sensing. And having in mind advanced works of this type my suggestion is to correct the title due to the lack of actual ET calculations from  spectral images according to own monitoring data. This is only a comparison of existing MODIS ETR results with traditional ones (Turc, Coutagne and Thornthwaite) and reading such a title the reader is sure that it is about pure conversions from spectral images, and it is not. Maybe cancelling of “…by MODIS Satellite Data….” would be satisfying (?).

It's a good idea of course to put together/correlate satellite ET products with simplified methods for ET/recharge estimations  and that should be appreciated for specific hydrogeological conditions, so you can find some advanced aspects here. Anyhow it could be treated as an innovative approach to this matter.

What is perhaps the most important conclusion that the close correspondence of actual evapotranspiration to the potential one exists, which appears as being dependent on the indicated factors typical for karst aquifers. Why not put this in conclusions?

Considering  equation p.6 l.190 Ar and Ae (AT AE??) lacks descriptions; AT and AE should have subscripts.

The figures are of good quality and legible, however, Fig 4 is unnecessary and should be omitted without affecting the understanding of the whole issue, just put a proper reference to MODIS in the text.

Usually ea is used as abbreviation for actual water vapor pressure; and es instead of esat;

Line 93 – Atm, osphere

Line 103 – space ….Moderate ,Resolution…

Line 486 – is form instead of from.

Line 202 – space …(1954)[35]

Line 202 – space …[41]and

The paper is clearly written, with sufficient explanation of the methods. The problem was solved and the adequate conclusions are presented.

All the references seem to be sufficient and cited in the paper, however the list should be supplemented with examples of the use of ET satellite data being calibrated on regional groundwater numerical models, like: Lubczynski M., Gurwin J., 2005: Integration of various data sources for transient groundwater modeling with spatio-temporally variable fluxes—Sardon study case, Spain. Journal of Hydrology 306 (1-4): 71-9.

So, please consider the suggestions given above

Round 2

Reviewer 1 Report

After looking carefully to the author's answers to my review my recommendation is to accept the paper as it is.